# LSB: Local Self-Balancing MCMC in Discrete Spaces

## Abstract

Markov Chain Monte Carlo (MCMC) methods are promising solutions to sample
from target distributions in high dimensions. While MCMC methods enjoy nice
theoretical properties, like guaranteed convergence and mixing to the true target, in
practice their sampling efficiency depends on the choice of the proposal distribution
and the target at hand. This work considers using machine learning to adapt the
proposal distribution to the target, in order to improve the sampling efficiency in
the purely discrete domain. Specifically, (i) it proposes a new parametrization for a
family of proposal distributions, called locally balanced proposals, (ii) it defines
an objective function based on mutual information and (iii) it devises a learning
procedure to adapt the parameters of the proposal to the target, thus achieving fast
convergence and fast mixing. We call the resulting sampler as the Locally Self-
Balancing Sampler (LSB). We show through experimental analysis on the Ising
model and Bayesian networks that LSB is indeed able to improve the efficiency
over a state-of-the-art sampler based on locally balanced proposals, thus reducing
the number of iterations required to converge, while achieving comparable mixing
performance.

## 1 Introduction

Sampling from complex and intractable probability distributions is of fundamental importance for
learning and inference [16]. MCMC algorithms are promising solutions to handle the intractability
of sampling in high dimensions and they have found numerous applications, in Bayesian statistics
and statistical physics [17, 23], bioinformatics and computational biology [3, 2] as well as machine
learning [4, 14, 18].

Although, MCMC can be applied to sample from any target distribution, in practice its efficiency
strongly depends on the choice of the proposal. Indeed, common phenomena, like slow convergence
and slow mixing, are typically the result of wrong choices of the proposal distribution. Therefore, it's
extremely important to devise strategies enabling the tuning of the proposal to target distributions [5,
13]. While there has been a lot of work focusing on designing machine learning-based strategies to
improve the efficiency of MCMC in the continuous domain [29, 20, 1, 6, 19], less effort has been
devoted to the discrete counterpart. Most common solutions consider continuous relaxations of
the problem by using embeddings and then leverage existing sampling strategies designed for the
continuous case. These strategies are suboptimal, either because they consider limited settings, where
the target distribution has specific analytic forms [29], or because they make strong assumptions on
the properties of the embeddings, thus not having guarantees of preserving the topological properties
of the original discrete domain [20, 1, 6, 19].

This work focuses on MCMC strategies for the purely discrete domain. Specifically, (i) we introduce
a new parametrization for a family of proposal distributions, called locally balanced proposals, which
have been recently studied in [28], (ii) we define an objective function based on mutual information,
which reduces the distance between the proposal and the target distribution and also reduces the

statistical dependence between consecutive samples, and (iii) we devise a learning procedure to adapt the parameters of the proposal to the target distribution using our objective. The resulting procedure, called the Local Self-Balancing sampler (LSB), automatically discovers an optimal locally balanced proposal, with the advantage of reducing the amount of user intervention and of improving the overall sampling efficiency, both in terms of convergence speed and mixing time.

We provide some empirical analysis of sampling from the 2D Ising model and from Bayesian networks and show that in some cases LSB is able to halve the number of iterations required to converge, while achieving similar mixing performance to [28].

We start by providing some background on locally balanced proposal distributions (Section 2), we introduce LSB by describing the parametrizations, the objective and the learning procedure (Section 3), we discuss the related work (Section 4) and the experiments (Section 5), and finally we conclude by highlighting the main limitations of LSB and possible future directions (Section 6).

## 2 Background

We consider the problem of sampling from a distribution $p$ with a support defined over a large finite discrete sample space $\mathcal{X}$, i.e. $p(\mathbf{x}) = \tilde{p}(\mathbf{x})/\sum_{\mathbf{x}'' \in \mathcal{X}} \tilde{p}(\mathbf{x}'')$, where the normalization term cannot be tractably computed and only $\tilde{p}$ can be evaluated. One solution to the problem consists of sampling using MCMC [17]. The main idea of MCMC is to sequentially sample from a tractable surrogate distribution, alternatively called proposal, and to use an acceptance criterion to ensure that generated samples are distributed according to the original distribution. More formally, MCMC is a Markov chain with a transition probability of the form:

$$T(\mathbf{x}'|\mathbf{x}) = A(\mathbf{x}', \mathbf{x})Q(\mathbf{x}'|\mathbf{x}) + 1[\mathbf{x}' = \mathbf{x}] \sum_{\mathbf{x}'' \in \mathcal{X}} \left(1 - A(\mathbf{x}'', \mathbf{x})\right)Q(\mathbf{x}''|\mathbf{x})) \tag{1}$$

where $Q(\mathbf{x}'|\mathbf{x})$ is the probability of sampling $\mathbf{x}'$ given a previously sampled $\mathbf{x}$, namely the proposal distribution, $1[\cdot]$ is the Kronecker delta function and $A(\mathbf{x}', \mathbf{x})$ is the probability of accepting sample $\mathbf{x}'$ given $\mathbf{x}$, e.g. $A(\mathbf{x}', \mathbf{x}) = \min\left\{1, \frac{\tilde{p}(\mathbf{x}')Q(\mathbf{x}|\mathbf{x}')}{\tilde{p}(\mathbf{x})Q(\mathbf{x}'|\mathbf{x})}\right\}$.[1] In this work, we consider the family of locally informed proposals [28], which are characterized by the following expression:

$$Q(\mathbf{x}'|\mathbf{x}) = \frac{g\left(\frac{\tilde{p}(\mathbf{x}')}{\tilde{p}(\mathbf{x})}\right)1[\mathbf{x}' \in N(\mathbf{x})]}{Z(\mathbf{x})} \tag{2}$$

where $N(\mathbf{x})$ is the neighborhood of $\mathbf{x}$ based on the Hamming metric.[2]

Note that the choice of $g$ has a dramatic impact on the performance of the Markov chain, as investigated in [28]. In fact, there is a family of functions called *balancing functions*, satisfying the relation $g(t) = tg(1/t)$ (for all $t > 0$), which have extremely desirable properties, briefly recalled hereunder.

**Acceptance rate.** The balancing property allows to rewrite the acceptance function in a simpler form, namely $A(\mathbf{x}', \mathbf{x}) = \min\left\{1, \frac{Z(\mathbf{x})}{Z(\mathbf{x}')}\right\}$. Therefore, a proper choice of $g$ can increase the ratio between the normalization constants $Z(\mathbf{x})$ and $Z(\mathbf{x}')$ with consequent increase of the acceptance rate even in high dimensional spaces.

**Detailed balance.** Note that for all $\mathbf{x}' = \mathbf{x}$, detailed balance trivially holds, viz. $p(\mathbf{x})T(\mathbf{x}'|\mathbf{x}) = p(\mathbf{x}')T(\mathbf{x}|\mathbf{x}')$. In all other cases, detailed balance can be proved, by exploiting the fact that $T(\mathbf{x}'|\mathbf{x}) = A(\mathbf{x}', \mathbf{x})Q(\mathbf{x}'|\mathbf{x})$ and by using the balancing property (see the Supplementary material for more details). Detailed balance is a sufficient condition for invariance. Consequently, the target $p$ is a fixed point of the Markov chain.

**Ergodicity.** Under mild assumptions, we have also ergodicity (we leave more detailed discussion to the Supplementary material). In other words, the Markov chain converges to the fixed point $p$ independently from its initialization.

---

[1]Other choices are available [17] as well.

[2]In other words, we consider all points having Hamming distance equal to 1 from $\mathbf{x}$.

80 **Efficiency.** The efficiency of MCMC is generally measured in terms of the resulting asymptotic
81 variance for sample mean estimators. This is indeed a proxy to quantify the level of correlation
82 between samples generated through MCMC. Higher levels of asymptotic variance correspond to
83 higher levels of correlation, meaning that the Markov chain produces more dependent samples and
84 it is therefore less efficient. Balancing functions are asymptotically optimal according to Peskun
85 ordering [28].

86 The work in [28] proposes a pool of balancing functions with closed-form expression together with
87 some general guidelines to choose one. However, this pool is only a subset of the whole family of
88 balancing functions and several cases do not even have an analytical expression. Consequently, it is
89 not clear which function to use in order to sample efficiently from the target distribution. Indeed, we
90 will see in the experimental section that (i) the optimality of the balancing function depends on the
91 target distribution and that (ii) in some cases the optimal balancing function may be different from the
92 ones proposed in [28]. In the next sections, we propose a strategy to automatically learn the balancing
93 function from the target distribution, thus achieving fast convergence (burn-in) and fast mixing.

## 94 3 LSB: Local Self-Balancing Strategy

95 We start by introducing two different parametrizations for the family of balancing functions in
96 increasing order of functional expressiveness. Then, we propose an objective criterion based on
97 mutual information, that allows us to learn the parametrization with fast convergence and fast mixing
98 on the target distribution.

### 99 3.1 Parametrizations

100 We state the following proposition and then use it to devise the first parametrization.

101 **Proposition 1.** *Given $n$ balancing functions $\mathbf{g}(t) = [g_1(t), \ldots, g_n(t)]^T$ and a vector of scalar*
102 *positive weights $\mathbf{w} = [w_1, \ldots, x_n]^T$, the linear combination $g(t) \doteq \mathbf{w}^T \mathbf{g}(t)$ satisfies the balancing*
103 *property.*

104 *Proof.* $g(t) = \mathbf{w}^T \mathbf{g}(t) = \sum_{i=1}^n w_i g_i(t) = t \sum_{i=1}^n w_i g_i(1/t) = t\mathbf{w}^T \mathbf{g}(1/t) = tg(1/t)$ $\qquad\square$

105 Despite its simplicity, the proposition has important implications. First of all, it allows to convert the
106 problem of choosing the optimal balancing function into a learning problem. Secondly, the linear
107 combination introduces only few parameters (in the experiments we consider $n = 4$) and therefore
108 the learning problem can be solved in an efficient way. The requirement about positive weights is
109 necessary to guarantee ergodicity (see Supplementary material on ergodicity for further details).

110 The first parametrization (LSB 1) consists of the relations $w_i = e^{\theta_i} / \sum_{j=1}^n e^{\theta_j}$ for all $i = 1, \ldots, n$,
111 where $\boldsymbol{\theta} = [\theta_1, \ldots, \theta_n] \in \mathbb{R}^n$. Note that the softmax is used to smoothly select one among the $n$
112 balancing functions. Therefore, we refer to this parametrization as learning to select among existing
113 balancing functions.

114 The second parametrization (LSB 2) is obtained from the following proposition.

115 **Proposition 2.** *Given $g_{\boldsymbol{\theta}}(t) = \min\{h_{\boldsymbol{\theta}}(t), th_{\boldsymbol{\theta}}(1/t)\}$, where $h_{\boldsymbol{\theta}}$ is a universal real valued function*
116 *approximator parameterized by vector $\boldsymbol{\theta} \in \mathbb{R}^k$ (e.g. a neural network), and any balancing function $\ell$,*
117 *there always exists $\tilde{\boldsymbol{\theta}} \in \mathbb{R}^k$ such that $g_{\tilde{\boldsymbol{\theta}}}(t) = \ell(t)$ for all $t > 0$.*

118 *Proof.* Given any balancing function $\ell$, we can always find a $\tilde{\boldsymbol{\theta}}$ such that $h_{\tilde{\boldsymbol{\theta}}}(t) = \ell(t)$ for all $t > 0$
119 (because $h_{\boldsymbol{\theta}}$ is a universal function approximator). This implies that $h_{\tilde{\boldsymbol{\theta}}}$ satisfies the balancing property,
120 i.e. $h_{\tilde{\boldsymbol{\theta}}}(t) = th_{\tilde{\boldsymbol{\theta}}}(1/t)$ for all $t > 0$. Consequently, by definition of $g_{\boldsymbol{\theta}}$, we have that $g_{\boldsymbol{\theta}}(t) = h_{\tilde{\boldsymbol{\theta}}}(t)$.
121 And finally we can conclude that $g_{\boldsymbol{\theta}}(t) = \ell(t)$ for all $t > 0$. $\qquad\square$

122 In theory, LSB 2 parameterizes the whole family of balancing functions and it allows to find the
123 optimal one from the whole set.[3] In practice, it is better to restrict the analysis only to monotonic
124 increasing functions, as we prefer to choose a proposal distribution sampling from regions of higher

---

[3]Note also that $g_{\boldsymbol{\theta}}(t)$ is a balancing function for any parameter $\boldsymbol{\theta}$.

125 probability mass. In the following proposition, we provide sufficient conditions to ensure the
126 monotonicity of LSB 2.

127 **Proposition 3.** *Define $g_{\boldsymbol{\theta}}$ as in LSB 2. Assume that $h_{\boldsymbol{\theta}}(t)$ is a differentiable and monotonic increasing*
128 *function with respect to variable $t$, satisfying the relation $h_{\boldsymbol{\theta}}(1/t) \geq \frac{1}{t}\frac{dh_{\boldsymbol{\theta}}(1/t)}{dt}$ for all $t > 0$. Then,*
129 *$g_{\boldsymbol{\theta}}$ is a monotonic increasing function in $t$.*

130 *Proof.* We have that $h_{\boldsymbol{\theta}}(1/t) - \frac{1}{t}\frac{dh_{\boldsymbol{\theta}}(1/t)}{dt} \geq 0$ for all $t > 0$. Also, note that $\frac{dth_{\boldsymbol{\theta}}(1/t)}{dt} =$
131 $h_{\boldsymbol{\theta}}(1/t) - \frac{1}{t}\frac{dh_{\boldsymbol{\theta}}(1/t)}{dt} > 0$. Therefore, $th_{\boldsymbol{\theta}}(1/t)$ is a monotonic increasing function in $t$ for
132 all $t > 0$. Now, consider any $t_1, t_2$ with $t_1 \geq t_2 > 0$, $g_{\boldsymbol{\theta}}(t_1) = \min\{h_{\boldsymbol{\theta}}(t_1), t_1 h_{\boldsymbol{\theta}}(1/t_1)\}$
133 and $g_{\boldsymbol{\theta}}(t_2) = \min\{h_{\boldsymbol{\theta}}(t_2), t_2 h_{\boldsymbol{\theta}}(1/t_2)\}$. By monotonicity of $h_{\boldsymbol{\theta}}(t)$ and $th_{\boldsymbol{\theta}}(1/t)$, we have that
134 $h_{\boldsymbol{\theta}}(t_1) \geq h_{\boldsymbol{\theta}}(t_2)$ and $t_1 h_{\boldsymbol{\theta}}(1/t_1) \geq t_2 h_{\boldsymbol{\theta}}(1/t_2)$. Therefore, we can conclude that $g_{\boldsymbol{\theta}}(t_1) \geq g_{\boldsymbol{\theta}}(t_2)$ for
135 all $t_1, t_2$ with $t_1 \geq t_2 > 0$. $\qquad\square$

136 Therefore, to build $g_{\boldsymbol{\theta}}$, we need a monotonic function $h_{\boldsymbol{\theta}}(t)$. Specifically, we can choose a monotonic
137 network [24] and constrain $h_{\boldsymbol{\theta}}$ to satisfy the condition $h_{\boldsymbol{\theta}}(1/t) \geq \frac{1}{t}\frac{dh_{\boldsymbol{\theta}}(1/t)}{dt}$ for all $t > 0$ (see the
138 Supplementary material for further information on how to impose the condition on $h_{\boldsymbol{\theta}}$). In the
139 next paragraphs, we propose an objective and a learning strategy to train the parameters of the two
140 parametrizations.

## 3.2 Objective and Learning Algorithm

142 The goal here is to devise a criterion to find the balancing function with the fastest speed of conver-
143 gence/mixing on the target distribution $p$. Note that the ideal case would be to sample from $p$ in an
144 independent fashion. We have already mentioned that this operation is computationally expensive
145 due to the intractability of computing the normalizing constant. In our case, we have to consider
146 the agnostic case, because the proposal distribution is a tractable surrogate for $p$. In this regard, we
147 define a criterion taking into account the distance from this ideal case. Specifically, we measure the
148 distance of the transition probability of the Markov chain in Eq. 1 from the target $p$ and the amount
149 of dependence between consecutive samples generated through it. In other words, we introduce the
150 following criterion, which is indeed a form of mutual information objective:

$$\mathcal{I}_{\boldsymbol{\theta}} = KL\big\{p(\boldsymbol{x})\tilde{T}_{\boldsymbol{\theta}}(\boldsymbol{x}'|\boldsymbol{x})\|p(\boldsymbol{x})p(\boldsymbol{x}')\big\} = E_p\big\{KL\{\tilde{T}_{\boldsymbol{\theta}}(\boldsymbol{x}'|\boldsymbol{x})\|p(\boldsymbol{x}')\}\big\} \qquad (3)$$

151 where $KL$ is the Kullback Leibler divergence and $E_p$ is the expected value of random vector $\boldsymbol{x}$
152 distributed according to $p$ and

$$\tilde{T}_{\boldsymbol{\theta}}(\boldsymbol{x}'|\boldsymbol{x}) \doteq \begin{cases} \frac{T_{\boldsymbol{\theta}}(\boldsymbol{x}'|\boldsymbol{x})}{Z_T} & \forall \boldsymbol{x}' \neq \boldsymbol{x} \\ 0 & \text{otherwise} \end{cases} \qquad (4)$$

153 is a conditional distribution defined over the transition probability $T_{\boldsymbol{\theta}}(\boldsymbol{x}'|\boldsymbol{x})$ in Eq. 1, where we have
154 explicited the dependence on $\boldsymbol{\theta}$ and we have introduced the normalizing constant $Z_T$, to ensure
155 that $\tilde{T}_{\boldsymbol{\theta}}(\boldsymbol{x}'|\boldsymbol{x})$ is a proper probability distribution. Note also that $\tilde{T}_{\boldsymbol{\theta}}(\boldsymbol{x}'|\boldsymbol{x})$ discards all pair of equal
156 samples, i.e. $\boldsymbol{x}' = \boldsymbol{x}$, as they are samples rejected by the Markov chain.

157 Minimizing Eq. 3 allows us to find the configuration of parameters bringing us "closer" in terms
158 of Kullback Leibler to the ideal case, namely $T_{\boldsymbol{\theta}}(\boldsymbol{x}'|\boldsymbol{x}) = p(\boldsymbol{x}')$ for all $\boldsymbol{x}' \neq \boldsymbol{x}$. The expectation in
159 Eq. 3 requires access to samples from $p$ and therefore cannot be computed. Nevertheless, note that
160 the KL term in Eq. 3 can be rewritten in an equivalent form (see the Supplementary material for the
161 derivation):

$$KL\{\tilde{T}_{\boldsymbol{\theta}}(\boldsymbol{x}'|\boldsymbol{x})\|p(\boldsymbol{x}')\} \propto \mathcal{J}(\boldsymbol{\theta},\boldsymbol{x}) \doteq E_{Q_{\boldsymbol{\theta}_0}}\Big\{\omega_{\boldsymbol{\theta},\boldsymbol{\theta}_0} A_{\boldsymbol{\theta}}(\boldsymbol{x}',\boldsymbol{x}) \log \frac{A_{\boldsymbol{\theta}}(\boldsymbol{x}',\boldsymbol{x})Q_{\boldsymbol{\theta}}(\boldsymbol{x}'|\boldsymbol{x})}{\tilde{p}(\boldsymbol{x}')}\Big\} \qquad (5)$$

162 where $\omega_{\boldsymbol{\theta},\boldsymbol{\theta}_0} = \frac{Q_{\boldsymbol{\theta}}(\boldsymbol{x}'|\boldsymbol{x})}{Q_{\boldsymbol{\theta}_0}(\boldsymbol{x}'|\boldsymbol{x})}$ and $\boldsymbol{\theta}_0$ is the reference parameter vector for the proposal distribution.
163 Alternatively to Eq. 3, we can minimize the following quantity:[4]

$$J(\boldsymbol{\theta}) = E_{Q_{init}}\{\mathcal{J}(\boldsymbol{\theta},\boldsymbol{x})\} + E_{Q_{\boldsymbol{\theta}_0}}\{\mathcal{J}(\boldsymbol{\theta},\boldsymbol{x})\} \qquad (6)$$

164 where $Q_{init}$ is the distribution used at initialization, typically uniform on the support $\mathcal{X}$. Note that
165 the first and the second terms in Eq. 6 encourage fast burn-in and fast mixing, respectively. Therefore,
166 $\theta$ can be learnt using the procedure described in Algorithm 1.

---

[4]See the Supplementary material for the modification of the objective for parametrization LSB 2.

**Algorithm 1:** Local Self-Balancing Training Procedure.

---

Learning rate $\eta = 1e - 4$, initial parameter $\boldsymbol{\theta}_0$, burn-in iterations $K$ and batch of samples $N$.

$\{\boldsymbol{x}_0^{(i)}\}_{i=1}^N \sim Q_{init}$ ;

**while** *k=1:K* **do**

    $\{\boldsymbol{x}_{init}^{(i)}\}_{i=1}^N \sim Q_{init}$ ;

    **while** *i=1:N* **do**

        $\widehat{\mathcal{J}}^{(i)\prime} \leftarrow$ Estimate of $\mathcal{J}(\boldsymbol{\theta}, \boldsymbol{x}_{init}^{(i)})$ using one sample from $Q_{\boldsymbol{\theta}_0}(\boldsymbol{x}|\boldsymbol{x}_{init}^{(i)})$ ;

        $\widehat{\mathcal{J}}^{(i)} \leftarrow$ Estimate of $\mathcal{J}(\boldsymbol{\theta}, \boldsymbol{x}_0^{(i)})$ using one sample from $Q_{\boldsymbol{\theta}_0}(\boldsymbol{x}|\boldsymbol{x}_0^{(i)})$ ;

    **end**

    $\widehat{J}(\boldsymbol{\theta}) = \frac{1}{N}\sum_{i=1}^N \widehat{\mathcal{J}}^{(i)\prime} + \frac{1}{N}\sum_{i=1}^N \widehat{\mathcal{J}}^{(i)}$ ;

    $\boldsymbol{\theta} \leftarrow \boldsymbol{\theta} - \frac{\eta}{N}\nabla_{\boldsymbol{\theta}}\widehat{J}(\boldsymbol{\theta})$ ;

    Update $\{\boldsymbol{x}_0^{(i)}\}_{i=1}^N$ with accepted samples ;

    $\boldsymbol{\theta}_0 \leftarrow \boldsymbol{\theta}$ ;

**end**

---

## 4   Related Work

It's important to devise strategies, which enable the automatic adaption of proposals to target distributions, not only to reduce user intervention, but also to increase the efficiency of MCMC samplers [5, 13]. Recently, there has been a surge of interest in using machine learning and in particular deep learning to learn proposals directly from data, especially in the continuous domain. Here, we provide a brief overview of recent integrations of machine learning and MCMC samplers according to different parametrizations and training objectives.

**Parametrizations and objectives in the continuous domain**. The work in [27] proposes a strategy based on block Gibbs sampling, where blocks are large motifs of the underlying probabilistic graphical structure. It parameterizes the conditional distributions of each block using mixture density networks and trains them using meta-learning on a log-likelihood-based objective. The work in [25] considers a global sampling strategy, where the proposal is parameterized by a deep generative model. The model is learnt through adversarial training, where a neural discriminator is used to detect whether or not generated samples are distributed according to the target distribution. Authors in [10] propose a global sampling strategy based on MCMC with auxiliary variables [12]. The proposals are modelled as Gaussian distributions parameterized by neural networks and are trained on a variational bound of a log-likelihood-based objective. The works in [15, 8] propose a gradient-based MCMC [7, 9], where neural models are used to learn the hyperparameters of the equations governing the dynamics of the sampler. Different objectives are used during training. In particular, the work in [8] uses a log-likelihood based objective, whereas the work in [15] considers the expected squared jump distance, namely a tractable proxy for the lag-1 autocorrelation function [21]. The work in [30] proposes a global two-stage strategy, which consists of (i) sampling according to a Gaussian proposal and (ii) updating its parameters using the first- and second-order statistics computed from a properly maintained pool of samples. The parameter update can be equivalently seen as finding the solution maximizing a log-likelihood function defined over the pool of samples. Finally, the work in [22] extends this last strategy to the case of Gaussian mixture proposals. All these works differ from the current one in at least two aspects. Firstly, it is not clear how these parametrizations can be applied to sampling in the discrete domain. Secondly, the proposed objectives compute either a distance between the proposal distribution and the target one, namely using an adversarial objective or a variational bound on the log-likelihood, or a proxy on the correlation between consecutive generated samples, namely the expected squared jump distance. Instead, our proposed objective is more general in the sense that it allows to (i) reduce the distance between the proposal and the target distribution as well as to (ii) reduce the statistical dependence between consecutive samples, as being closely related to mutual information.

**Sampling in the discrete domain**. Less efforts have been devoted to devise sampling strategies for a purely discrete domain. Most of the works consider problem relaxations by embedding the discrete domain into a continuous one, applying existing strategies like Hamiltonian Monte Carlo [29, 20, 1, 6, 19] on it and then moving back to the original domain. These strategies are

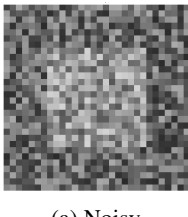
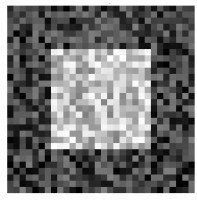

(a) Noisy                                        (b) Clean

Figure 1: Examples of $\boldsymbol{\alpha}$ in different settings of the Ising model ($30\times 30$), i.e noisy $\mu = 1, \sigma = 3$ and clean $\mu = 3, \sigma = 3$.

suboptimal, either because they consider limited settings, where the target distribution has specific analytic forms [29], or because they make strong assumptions on the properties of the embeddings, thus not preserving the topological properties of the discrete domain [20, 1, 6, 19].[5]. The work in [28] provides an extensive experimental comparison between several discrete sampling strategies, including the ones based on embeddings, based on stochastic local search [11] and the Hamming ball sampler [26], which can be regarded as a more efficient version of block Gibbs sampling. Notably, the sampling strategy based on locally informed proposals and balancing functions proposed in [28] can be considered as the current state of the art for discrete MCMC. Our work builds and extends upon this sampler by integrating it with a machine learning strategy. To the best of our knowledge, this is the first attempt to consider the integration of machine learning and MCMC in the discrete setting.

# 5    Experiments

Firstly, we analyze samplers' performance on the 2D Ising model. Then, we perform experiments on additional UAI benchmarks. Code to replicate the experiments in this section is available in the Supplementary material. All experiments are performed on a laptop provided with 4 Intel i5 cores (2 GHz) and 16 GB of RAM memory.

## 5.1    2D Ising Model

The Ising model has been introduced in statistical mechanics in 1920 and it has been applied in several domains since then. In this section, we consider an application to image analysis, where the goal is to segment an image to identify an object from its background. Consider a binary state space $\mathcal{X} = \{-1, 1\}^V$, where $(V, E)$ defines a square lattice graph of the same size of the analyzed image, namely $n \times n$. For each state configuration $\mathbf{x} = (x_i)_{i\in V} \in \mathcal{X}$, define a prior distribution

$$p_{prior}(\mathbf{x}) \propto \exp\left\{ \lambda \sum_{(i,j)\in E} x_i x_j \right\}$$

where $\lambda$ is a non-negative scalar used to weight the dependence among neighboring variables in the lattice. Then, consider that each pixel $y_i$ is influenced only by the corresponding hidden variable $x_i$ and generated according to a Gaussian density with mean $\mu x_i$ and variance $\sigma^2$. Note that each variable in the lattice tells whether the corresponding pixel belongs to the object or to the background (1 or -1, respectively). The corresponding posterior distribution of a hidden state $\mathbf{x}$ given an observed image is defined as follows:

$$p(\mathbf{x}) = \frac{1}{Z} \exp\left\{ \sum_{i\in V} \alpha_i x_i + \lambda \sum_{(i,j)\in E} x_i x_j \right\} \tag{7}$$

where $\alpha_i = y_i \mu / \sigma^2$ is a coefficient biasing $x_i$ towards either 1 or $-1$. Therefore, $\boldsymbol{\alpha} = (\alpha_i)_{i\in V}$ contains information about the observed image. Figure 1 shows two synthetically generated examples of $\boldsymbol{\alpha}$. Our goal is to analyze the sampling performance on the distribution defined in Eq. 7.

**Universally optimal balancing function.**    We start by comparing the balancing functions proposed in [28], namely $g(t) = t/(1+t)$ (a.k.a Barker function), $\sqrt{t}, \min\{1, t\}$ and $\max\{1, t\}$, in order to test

---

[5]For example by considering transformations that are bijective and/or by proposing transformations which allow to tractably compute the marginal distribution on the continuous domain.

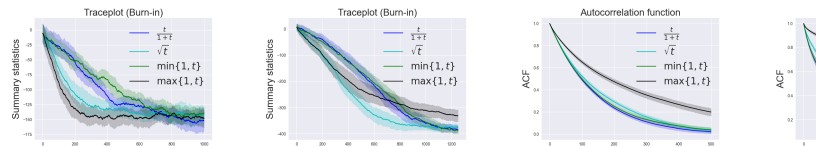

(a) Independent (burn-in)  (b) Dependent (burn-in)  (c) Independent (mixing)  (d) Dependent (mixing)

Figure 2: Samplers' performance on noisy and clean cases of the Ising model ($30 \times 30$). (a)-(b) are the traceplots for the burn-in phase, (c)-(d) are the autocorrelation functions for the mixing one.

Table 1: Quantitative performance for mixing measured by effective sample size on the noisy and clean cases of the Ising model ($30 \times 30$). $\max\{1, t\}$ is performing significantly worse in statistical terms than the other functions.

| Setting | $\frac{t}{1+t}$ | $\sqrt{t}$ | $\min\{1, t\}$ | $\max\{1, t\}$ |
|---|---|---|---|---|
| Noisy | $\mathbf{2.48 \pm 0.21}$ | $\mathbf{2.30 \pm 0.22}$ | $\mathbf{2.42 \pm 0.19}$ | $1.75 \pm 0.17$ |
| Clean | $\mathbf{2.58 \pm 0.73}$ | $\mathbf{1.99 \pm 0.43}$ | $\mathbf{2.56 \pm 0.62}$ | $1.26 \pm 0.12$ |

whether there is a universally optimal balancing function among this subset. In particular, we run the samplers over two instances of the Ising model, viz. a setting with independent $(\lambda, \mu, \sigma) = (0, 1, 3)$ and another one with dependent $(\lambda, \mu, \sigma) = (1, 1, 3)$ variables. We evaluate the performance over 30 repeated trials by analyzing the convergence speed during the burn-in phase (using traceplots) and the mixing time (computing the autocorrelation function and the effective sample size). We visualize the corresponding results in Figure 2 and report the quantitative performance in Table 1. Further details about the simulations are available in the Supplementary Material.

By comparing the performance of convergence speed and mixing time in Figure 2, we observe that unbounded functions, like $\max\{1, t\}$ and $\sqrt{t}$, tend to converge faster while having slower mixing compared to the other two functions, thus being in line with the empirical findings of [28]. This is due to the fact that unbounded functions have an intrinsic preference for visiting more likely regions at the cost of a reduced amount of exploration. Moving a step further, we compare Figure 2a and Figure 2b and observe that the optimal function is different for the two cases (i.e. $\max\{1, t\}$ in the independent case and $\sqrt{t}$ in the dependent one). These results suggest that optimality not only depends on the performance of burn-in and mixing but also on the distribution we are sampling from. This allows us to reject the hypothesis about the existence of a universal optimum among the pool of balancing functions proposed in [28] and to motivate our next set of experiments, where the aim is to learn to adapt the balancing function to the target distribution.

**Learning the balancing function.** We compare the four balancing functions used in the previous set of experiments with our two parametrizations on four different settings of the Ising model, namely independent and noisy $(\lambda, \mu, \sigma) = (0, 1, 3)$, independent and clean $(\lambda, \mu, \sigma) = (0, 3, 3)$, dependent and noisy $(\lambda, \mu, \sigma) = (1, 1, 3)$ and dependent and clean $(\lambda, \mu, \sigma) = (1, 3, 3)$ cases and show the corresponding performance in Figure 3 and Table 2. We leave additional details and results to the Supplementary Material.

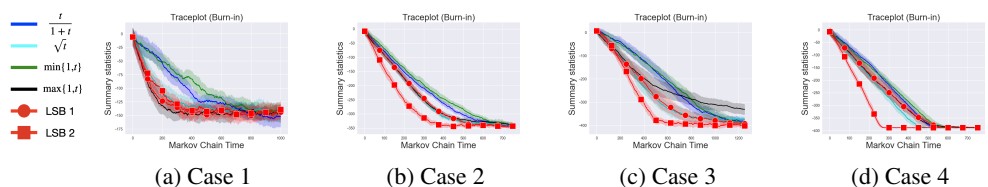

(a) Case 1  (b) Case 2  (c) Case 3  (d) Case 4

Figure 3: Samplers' performance on four cases of the Ising model ($30 \times 30$) for the burn-in phase. (a) Case 1: Independent-noisy, (b) case 2: Independent-clean, (c) case 3: Dependent-noisy, (d) case 4: Dependent-clean

Table 2: Quantitative performance for mixing measured by effective sample size on the four cases of the Ising model ($30 \times 30$). $\max\{1, t\}$ is performing significantly worse in statistical terms than the other functions.

| Setting | $\frac{t}{1+t}$ | $\sqrt{t}$ | $\min\{1, t\}$ | $\max\{1, t\}$ | LSB 1 | LSB 2 |
|---|---|---|---|---|---|---|
| Case 1 | $2.48 \pm 0.21$ | $2.30 \pm 0.22$ | $2.42 \pm 0.19$ | $1.75 \pm 0.17$ | $2.50 \pm 0.28$ | $2.46 \pm 0.28$ |
| Case 2 | $3.33 \pm 0.32$ | $2.94 \pm 0.36$ | $3.33 \pm 0.33$ | $1.72 \pm 0.18$ | $2.98 \pm 0.24$ | $3.33 \pm 0.43$ |
| Case 3 | $2.58 \pm 0.73$ | $1.99 \pm 0.43$ | $2.56 \pm 0.62$ | $1.26 \pm 0.12$ | $2.48 \pm 0.61$ | $2.67 \pm 0.84$ |
| Case 4 | $32.8 \pm 9.2$ | $18.5 \pm 6.8$ | $31.8 \pm 10.0$ | $2.60 \pm 1.46$ | $18.4 \pm 8.0$ | $30.8 \pm 9.2$ |

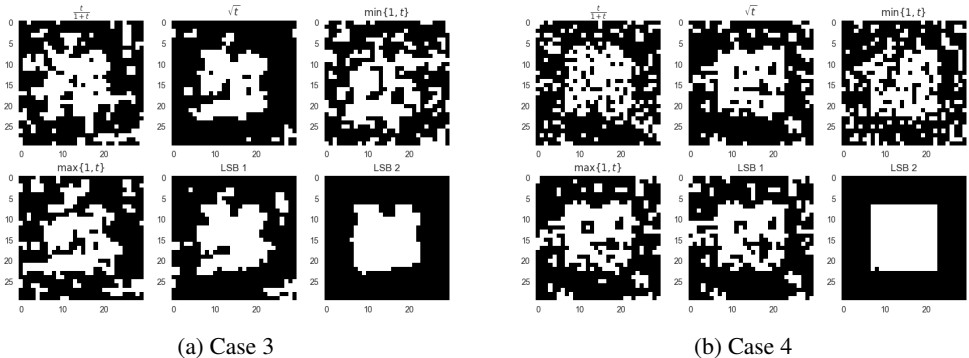

(a) Case 3                    (b) Case 4

Figure 4: Realizations obtained after 500 (Case 3) and 300 (Case 4) burn-in iterations on the Ising model.

From Figure 3, we can see that our first parametrization LSB 1 is able to always "select" an unbounded balancing function during burn-in, while when approaching convergence it is able to adapt to preserve fast mixing, as measured by the effective sample size in Table 2. It's interesting to mention also that the softmax nonlinearity used in LSB 1 can sometimes slow down the adaptation due to vanishing gradients. This can be observed by looking at the case 4 of Figure 3, where for a large part of the burn-in period the strategy prefers $\max\{1, t\}$ over $\sqrt{t}$. Nevertheless, it is still able to recover a solution different from $\max\{1, t\}$ at the end of burn-in, as confirmed by the larger effective sample size in Table 2 compared to the one achieved by $\max\{1, t\}$.

Furthermore, we observe that our second parametrization LSB 2, which is functionally more expressive compared to LSB 1, allows to outperform all previous cases in terms of convergence speed, while preserving optimal mixing, as shown in Figure 3 and Table 2. This provides further evidence that the optimality of the balancing function is influenced by the target distribution and that exploiting such information can dramatically boost the sampling performance (e.g. in case 3 of Figure 3, LSB 2 converges twice time faster as the best balancing function $\sqrt{t}$). Figure 4 provides some realizations obtained by the samplers for the cases with dependent variables $\lambda = 1$. We clearly see from these pictures that convergence for LSB 2 occurs at an earlier stage than the other balancing functions and therefore the latent variables in the Ising model converge faster to their ground truth configuration.

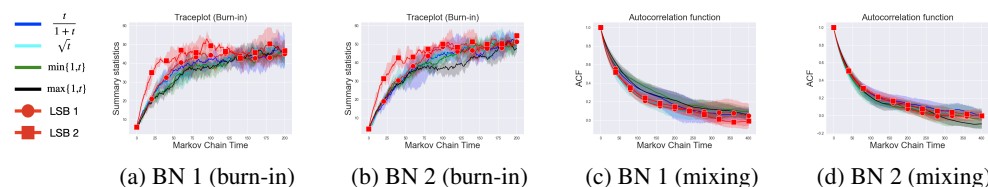

(a) BN 1 (burn-in)      (b) BN 2 (burn-in)      (c) BN 1 (mixing)      (d) BN 2 (mixing)

Figure 5: Samplers' performance on Bayesian networks from UAI competition (100 variables). (a)-(b) are the traceplots for the burn-in phase, (c)-(d) are the autocorrelation functions for the mixing one.

Table 3: Quantitative performance for mixing measured by effective sample size on two Bayesian networks from UAI competition.

| Dataset | $\frac{t}{1+t}$ | $\sqrt{t}$ | $\min\{1,t\}$ | $\max\{1,t\}$ | LSB 1 | LSB 2 |
|---------|------|------|------|------|------|------|
| BN 1 | $2.90 \pm 0.76$ | $3.41 \pm 0.77$ | $2.54 \pm 0.32$ | $2.70 \pm 0.63$ | $3.19 \pm 0.46$ | $3.22 \pm 0.38$ |
| BN 2 | $3.43 \pm 0.75$ | $3.92 \pm 0.94$ | $3.78 \pm 0.50$ | $3.63 \pm 0.67$ | $3.52 \pm 0.42$ | $3.44 \pm 0.44$ |

## 5.2 Bayesian Networks: UAI data

We evaluate how our strategy generalizes to different graph topologies compared to the one of the Ising model. In particular, we consider two Bayesian networks, with 100 discrete variables each and near-deterministic dependencies, from the 2006 UAI competition.[6] Similarly to the previous experiments for the Ising model, we measure the performance over 5 repeated trials by analyzing the convergence speed during the burn-in phase (using traceplots) and the mixing time (computing the autocorrelation function and the effective sample size). Further details about the simulations are available in the Supplementary Material.

We observe that the proposed strategy is able to adapt to the target distribution, thus achieving fast convergence (Figure 5) while preserving fast mixing (both Figure 5 and Table 3) compared to existing balancing functions.

## 6 Conclusion

We have presented a strategy to learn locally informed proposals for MCMC in discrete spaces. The strategy consists of (i) a new parametrization of balancing functions and (ii) a learning procedure adapting the proposal to the target distribution, in order to improve the sampling performance, both in terms of convergence speed and mixing.

Note that the LSB sampler belongs to the family of local sampling strategies, thus inheriting their limitations. The locality assumption can be quite restrictive, for example when sampling from discrete distributions with deterministic dependencies among variables. In such situations, local sampling might fail to correctly sample from the target in a finite amount of time, as being required to cross regions with zero probability mass. This remains an open challenge to be investigated in future work.

It's important to mention that this work is foundational and general. The proposed strategy reduces the amount of user intervention and improves the sampling efficiency. However, these results could have potential impacts on society. At first glance, one could argue that the automation of the sampling procedure could have a negative impact on society as reducing the amount of human labour required to run the sampling strategy. However, we think that the benefits are of a far greater number compared to the negative aspects. In fact, the procedure reduces the costs of domain knowledge, with the advantage of democratizing the sampling strategy and reducing suboptimal configurations of the algorithm resulting from possibly wrong human decisions. Furthermore, the proposed strategy introduces a small amount of additional computation, which is used to reduce the amount of queries to the target distribution, thus improving the query and the sampling efficiency. We think that this last aspect could contribute positively towards devising more energy-efficient algorithms and therefore being more environment friendly.

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
