# OpenReview forum: "LSB: Local Self-Balancing MCMC in Discrete Spaces"
_NeurIPS.cc/2021/Conference — NeurIPS 2021 Submitted_

### Official Review · Reviewer_ufwd · 2021-07-10

**Rating:** 5
**Confidence:** 4

**Summary:**

The authors propose a method to adapt the MCMC proposal to the discrete distribution of interest. The parametric proposal is constructed to be locally balanced, and the parameters are learned by maximizing a mutual information-based objective.

**Limitations And Societal Impact:**

Yes.

**Main Review:**

The paper is well-written and I did not have problems following the exposition.

The main idea of the paper is largely based on previous work by Zanella [1] on locally-balanced proposals for MCMC on discrete distributions. Rather than hardcoding such a proposal, the authors propose learning a parametric form either as a linear combination of known proposals or as a (monotonic) neural network. The setup of the optimization problem in eq. (3) to learn these parametric proposal could have been better motivated in the paper. More importantly, the transition from eq. (3) to eq. (6), which is the actual objective optimized, seems rather heuristic and not fully explained. Is there an intuitive reason why it may not be a problem to completely discard the expectation over $p$ in eq. (3)?

The experiments on some small-scale Ising models and on UAI Bayes nets show that the proposed methods seems to be comparable or faster in terms of burn-in and mixing. However, the authors do not show plots comparing the actual (wall-clock) time of the different sampling methods. The original paper by Zanella [1] already comments on the fact that locally-balanced proposals incur computational overhead (e.g., in comparison to a M-H sampler), and I assume that the steps in algorithm 1 for computing and optimizing the objective $J$ increase this overhead. As a result, it is hard to judge whether the proposed method is actually superior in practice.

Furthermore, at least for the Ising model simulations, the authors could compare to a recently proposed method by Grathwolh et al. [2], which use gradient-based local information in the proposal.

Other comments:
* The plots in Fig. 2, 3, and 5 are too small and hard to read.
* It is not explained what exactly the summary statistics are in the plots that evaluate the burn-in.

&nbsp;
### References

[1] Zanella. Informed proposals for local MCMC in discrete spaces. J. Am. Stat. Assoc. '20.\
[2] Grathwohl et al. Oops I Took A Gradient: Scalable Sampling for Discrete Distributions. ICML '21.

**Time Spent Reviewing:**

5

---

> ### Author Response · Authors · 2021-08-10
> **Clarifications about Objective and Comparison with Competitors**
>
> We thank the reviewer for the thoughtful comments. Please, find below the answers to your questions.
>
> **Clarifications about objective**
>
> Our objective attempts to minimize the dependence between consecutive samples, by minimizing an approximation of Eq. (3). Indeed,  we approximate $p$ using a mixture between a uniform distribution, namely $Q_{init}$, and a more informative one, namely $Q_{\theta_0}$. The objective in Eq. (6) can be derived by the linearity of expectation on this mixture. Note that discarding $p$ would be equivalent to approximate $p$ using a uniform distribution, namely using only $Q_{init}$, and we would end up to compute the KL divergence between the transition probability and the true target, thus discarding the dependence about consecutive samples (as measured by mutual information). However, we can do better than that by exploiting also the term with $Q_{\theta_0}$, as being more informative than $Q_{init}$.
>
> Also, note that a similar objective has been proposed in [1] (see the corresponding Eq. (8) in [1]) to automatically configure the hyperparameters of a continuous MCMC sampler. However, the authors in [1] use the squared jump distance in place of our KL. In our case, the squared jump distance is uninformative (recall that in our discrete case the neighborhood is based on the Hamming distance 1 and therefore the squared jump distance is constant for all possible steps). The connection with [1] highlights another important interpretation of our objective. Indeed, the first addend in our Eq. (6) (equivalently the second addend in Eq. (8) of [1]) encourages fast burn-in, while the second addend in our Eq. (6) encourages fast mixing.
>
>
> **Comparison with [0]**
>
> We agree with you that our strategy requires to compute and optimize the objective $J$, but this is done only during the burn-in phase and the optimization is performed over a relatively small amount of parameters (we have 4 parameters for LSB 1 and 800 parameters for LSB 2). During mixing, we only have a small amount of additional cost with respect to [0] due to the computation of the parameterized balancing functions. In any case, we can discuss the computational overhead introduced by our strategy and add the wall-clock times in the tables.
>
> Apart from the computational aspect, there is a clear advantage in terms of sampling efficiency with respect to [0]. Note that we can recover balancing functions requiring a smaller number of burn-in iterations and at the same time having good mixing behaviour (see for example Figure 3 and 4 together with Table 2). This is not the case for [0], as the proposed balancing functions are either good in terms of burn-in or good in terms of mixing (see Figure 2). Overall, the number of evaluations of the target distribution is reduced compared to [0] and we do not need to specify which balancing function to use.
>
>
> **Comparison with [1]**
>
> We feel it’s quite unfair to ask for an experimental comparison against such recent work. However, we can provide a detailed discussion which clearly highlights the novelties of our strategy.
>
> Let us provide some technical details and summarize everything in a table. First of all, note that both works ([1] and ours) can be seen as an extension of [0]. In fact, if we consider a target distribution of the form $p(x)=\frac{e^{f(x)}}{Z}$, where $x\in\\{0,1\\}^d$ and that all works use a neighborhood based on a Hamming distance of 1, all proposals boil down to the following simple form:
> $$Q(i|x)=Categorical\frac{g(e^{d(x)_i})}{\sum_j g(e^{d(x)_j})}$$
> where Q(i|x) is the probability of flipping the $i$-th bit in $x$ (and we refer to $x_i$ as the corresponding flipped vector). Therefore, we can summarize the main properties of all strategies in the following table:
>
> | Name  | Balancing function $g(\cdot)$  | Local difference $d(x)_i$  | assumptions on $p(x)$ (in addition to [0]) | Target evaluations per iteration  |
> |:---|---|---|---|---|
> | [0]  | Set manually  | Exact $d(x)_i=f(x_i)-f(x)$  | - | O(d)  |
> | [1] | Set manually, i.e. $g(\cdot)=\sqrt{\cdot}$  | Approximate (Taylor approximation of $d(x)_i$)  | $f$ is known, differentiable and $d(x)_i$ is L-Lipschitz  | O(1)  |
> | Ours  | Learnt  | Exact $d(x)_i=f(x_i)-f(x)$  | - | O(d)  |
>
> Note that [1] exploits the additional assumptions on $p(x)$ and uses gradient information to approximate $d(x)_i$, thus allowing to reduce the number of target evaluations per iteration compared to [0] (and consequently also to our work). However, our goal is different from [1], as we want to identify the balancing function requiring a small number of burn-in iterations and achieving fast mixing in an automated fashion, thus avoiding manual configuration. In [1], the balancing function is fixed ($\sqrt(\cdot)$) and not necessarily optimal (see for example Figure 2c and 2d or Figure 4a and 4b in our experiments).
> Also, the two methods ([1] and ours) are orthogonal to each other and nothing prevents us on applying the same approximation of [1] to our setting, to further reduce the overall number of target evaluations (per iteration thanks to [1] and globally thanks to the learnt balancing function). However, it’s important to mention that there are cases in which the approximation of [1] cannot be used, for example in the case of non-differentiable target distributions or distributions with no analytical formula (e.g. UAI data).
>
>
> **Minor comments**
>
> We can increase the fonts of the figures and explain what summary statistics are (i.e. $\sum_i x_i$).
>
> [0] “Informed proposals for local MCMC in discrete spaces” by Zanella 2020.
>
> [1] "Oops I Took A Gradient: Scalable Sampling for Discrete Distributions" by Gratwohl et.al 2021

---

### Official Review · Reviewer_ueJZ · 2021-07-11

**Rating:** 3
**Confidence:** 5

**Summary:**

This work proposed a strategy to learn locally informed proposals for MCMC in discrete spaces. In spirit, it is more or less similar to the multiple-try Metropolis method proposed in Liu et al. (2001), and it is unclear whether the new method is more efficient than the multiple-try Metropolis method.

Liu, J. S., Liang, F. and Wong, W. H. (2000). The multiple-try method and local optimization in Metropolis sampling, Journal of the American Statistical Association, 95(449): 121–134.

**Main Review:**

This work presents a new parametrization for learning locally balanced proposals, and it is basically can be viewed as  a supplementary material of the recent work Zanella (2020).

] G. Zanella. Informed Proposals for Local MCMC in Discrete Spaces.  Journal of the American Statistical Association, 115(530):852–865, 2020.

**Time Spent Reviewing:**

5

---

> ### Author Response · Authors · 2021-08-09
> **Clarifications**
>
> We thank the reviewer for the time spent to read the paper.
>
> We respectfully disagree with the provided comments. See below for some clarifications and questions:
> 1. “More or less similar to Multiple-try Metropolis” (MTM). Can you please be precise? Firstly, note that our work is based on locally balanced proposals (Zanella 2020) and there is no equivalent of a balancing function in MTM (see discussion and comparison, in the paper of Zanella). Secondly, our aim is to automate the selection of the balancing function, thus building upon Zanella’s work.
> 2. “It can be viewed as a supplementary material of the recent work of Zanella”. On which basis, can the reviewer state this? We are proposing new parametrizations for the family of balancing functions and a strategy to learn them, thus automating the process of configuration of the sampler. Furthermore, the strategy discovers balancing functions, which requires a small number of burn-in iterations and achieve similar mixing performance compared to the best performing balancing ones.

---

### Official Review · Reviewer_2Vii · 2021-07-12

**Rating:** 5
**Confidence:** 4

**Summary:**

The paper address the problem of sampling in discrete spaces by learning proposal distributions that could potentially lead to faster convergence. The paper primarily builds on the original idea by G.Zanella, 2020 where the idea of locally balanced proposals was first proposed. The authors in this work aim to learn these proposals for "better" sampling in discrete spaces.

**Limitations And Societal Impact:**

Yes, the societal impact is adequately discussed.

**Main Review:**

**Positives:** The paper is well written and easy to follow. Sampling in discrete spaces is a challenging problem and the paper proposed a nice solution to learn proposal distributions for this. As discussed in the paper, the idea address to a large extent the limitations of previous works of Afshar and Domke, 2015, Pakman and Paninski 2013, Nishimura et.al 2020 etc.

**Limitations and Questions:**

1) **Prior work:** The authors missed two recent work on sampling in discrete spaces that also learn proposal distributions namely "Sampling in Combinatorial Spaces using SurVAE Flow Augmented MCMC" by Jaini et.al 2021 where the authors use normalizing flows to learn a transformation to transform the discrete distribution to a "easy to sample" continuous distribution. This is in-line with the other work on sampling in discrete spaces by using continuous relaxations but the paper also address the drawbacks of the previous works as mentioned by the authors and "Oops I Took A Gradient: Scalable Sampling for Discrete Distributions" by Gratwohl et.al 2021 where the authors also use the idea of locally balanced proposals for scalable sampling in discrete spaces. I believe at the very least these papers need to discussed as they are directly relevant to the problem and solution the authors propose. Finally, the idea of learning proposal distributions (atleast in continuous spaces) has been explored in a lot of detail by [Marzouk and Spantini] and [Hoffman et.al 2019, Neutralizing bad geometry in HMC using Neural transport] which I believe is also relevant to this work and warrants a discussion.

2) **Use of Universal Approximation to build locally balanced proposals:** The authors appeal to the universal approximation result to construct the family of proposals named LSB 2. While, theoretically, this affords advantages to the LSB 2 sampler,  the approximation results necessarily requires an infinite capacity network for such guarantees.  In practice, since finite capacity networks are used for function approximation (and thus the resultant locally balanced proposals) this would incur some error since the approximation will never be exact. I wonder if the sampler suffers due to this error and if the resultant proposal is still locally balanced. Put another way, since h_{\theta} is of limited capacity in practice, it approximated a locally balanced proposal function g within some error epsilon. Does this error break the properties of locally-balanced proposals and does all the theoretical advantages still hold? What are the implications of this error on the sampler (eq 2. in particular).

3) **Experiments:** I believe the paper will benefit greatly from a more detailed experimental evaluation that involves comparisons to other methods. Specifically, I believe that apart from the present experiments where the comparison is mostly made wrt other locally balanced proposals, the authors should also compare to other samplers including discrete MCMC (random-walk), Gibbs sampling, discrete HMC (Nishimura et.al, code is available publicly), SurVAE+MH (Jaini et.al, code is public), and Gratwohl et.al (code is public) to back up claims of faster convergence and better performance. I also think that it will be interesting to see how this idea of learning locally balanced proposals work in higher dimensional problems eg. Ising model for denoising MNIST, or quantized bayesian neural nets.

**Time Spent Reviewing:**

5.5

---

> ### Author Response · Authors · 2021-08-09
> **Discussion of prior work and clarifications**
>
> Thank you for the appreciation of our work (positive aspects) and for pointing out two recent and related works, which allow us to better highlight our novelties. Please see below the clarifications to your questions:
>
> **Prior Work**
>
> We were not aware about the works [1] and [2] at the time of submission. In any case, we are grateful for the pointers and discuss these works hereunder.
>
> *Similarities and differences with respect to [1]*
>
> Let us provide some technical details to better highlight the relations and summarize everything in a table. First of all, note that both works ([1] and ours) can be seen as an extension of [0]. In fact, if we consider a target distribution of the form $p(x)=\frac{e^{f(x)}}{Z}$, where $x\in\\{0,1\\}^d$ and that all works use a neighborhood based on a Hamming distance of 1, all proposals boil down to the following simple form:
> $$Q(i|x)=Categorical\frac{g(e^{d(x)_i})}{\sum_j g(e^{d(x)_j})}$$
> where Q(i|x) is the probability of flipping the $i$-th bit in $x$ (and we refer to $x_i$ as the corresponding flipped vector). Therefore, we can summarize the main properties of all strategies in the following table:
>
> | Name  | Balancing function $g(\cdot)$  | Local difference $d(x)_i$  | assumptions on $p(x)$ (in addition to [0]) | Target evaluations per iteration  |
> |:---|---|---|---|---|
> | [0]  | Set manually  | Exact $d(x)_i=f(x_i)-f(x)$  | - | O(d)  |
> | [1] | Set manually, i.e. $g(\cdot)=\sqrt{\cdot}$  | Approximate (Taylor approximation of $d(x)_i$)  | $f$ is known, differentiable and $d(x)_i$ is L-Lipschitz  | O(1)  |
> | Ours  | Learnt  | Exact $d(x)_i=f(x_i)-f(x)$  | - | O(d)  |
>
> Note that [1] exploits the additional assumptions on $p(x)$ and uses gradient information to approximate $d(x)_i$, thus allowing to reduce the number of target evaluations per iteration compared to [0] (and consequently also to our work). However, our goal is different from [1], as we want to identify the balancing function requiring a small number of burn-in iterations and achieving fast mixing in an automated fashion, thus avoiding manual configuration. In [1], the balancing function is fixed and not necessarily optimal (as we show in the experiments, see for example Figure 2c and 2d or Figure 4a and 4b).
> Also, the two methods ([1] and ours) are orthogonal to each other and nothing prevents us on applying the same approximation of [1] to our setting, to improve the overall computational efficiency. However, it’s important to mention that there are cases in which the approximation of [1] cannot be used, for example in the case of non-differentiable target distributions or distributions with no analytical formula (e.g. UAI data).
>
> *Similarities and differences with respect to [2]*
>
> Both works learn a proposal distribution. However, [2] falls in the class of continuous relaxation methods, while our strategy does not learn an embedding.  From a computational point of view, note that our strategy parametrizes a one-dimensional real function, thus using a much smaller number of parameters (in case of LSB 1 we have only 4 parameters and in case of LSB 2 we have 800 parameters) compared to [2]. Also, the number of parameters does not depend on dimension $d$, differently from [2], which can require larger networks to transform higher dimensional raw data.
>
> **Use of Universal Approximation to build locally balanced proposals**
>
> No, the finite capacity of the network does not affect the balancing property. Indeed, $g_{\theta}(t)$ is always a balancing function by definition, independently from the capacity of $h_{\theta}(t)$ (recall that $g_{\theta}(t)=\min\\{h_{\theta}(t),t h_{\theta}(1/t)\\}$ and you can apply the balancing property to verify that the statement holds). However, the finite capacity affects the possible balancing functions we can represent. In our experiments, we found that the proposed network is enough to approximate well the known balancing functions (at least three out of the four ones proposed in [0]) and to achieve good performance, but it is not able to learn balancing functions like $\max\\{1,t\\}$ (See Figure 4 in the Supplementary). Increasing the network capacity allows to increase the subset of balancing functions that you can represent at the expense of a larger number of parameters and a higher computational overhead.
>
> **Experiments**
>
> We feel it’s quite unfair to ask for a comparison with so recent works like [1] and [2]. However, we think that the above discussion addresses your questions and highlights the novelties.
>
> [0] “Informed proposals for local MCMC in discrete spaces” Zanella 2020.
>
> [1] "Oops I Took A Gradient: Scalable Sampling for Discrete Distributions" by Gratwohl et.al 2021
>
> [2] "Sampling in Combinatorial Spaces using SurVAE Flow Augmented MCMC" by Jaini et.al 2021

---

> > ### Comment · Reviewer_2Vii · 2021-08-23
> > **Questions follwoing author's response**
> >
> > I thank the authors for the detailed replies. I still have a couple of points following the authors' reply:
> >
> > **Prior work**: The authors very nicely summarized the differences of their work with [1] and [2] and I hope they would include this discussion in the revision for sake of completeness wrt prior work. However, a main aim of the paper, as the authors themselves also stress, is to automatically identify local balancing functions for short burn-in and faster mixing. While there are examples in the paper that show this for a choice of few balancing functions, I think at the very least this needs to be compared with the alternatives in [1] and [2] to weigh the advantages the proposed method affords in comparison since the underlying problem is that of sampling in discrete spaces and ultimately the interest is to identify an efficient sampler.
> >
> > I also do not agree with the authors that it unfair to ask for comparison with [1] and [2]. Both the papers were released around four months before the NeurIPS deadline (first week of February). Additionally, both the papers have publicly available code that is fairly easy to use and quick to setup. Finally, both the papers have experiments on a diverse set of problems comparing to several alternatives which I believe is required to judge the effectiveness of a sampler.
> >
> > **Universal Approximation**: The authors suggested that the finite capacity does not effect the local balancing property but I am not clear on that yet.
> >
> > From my understanding $g(\cdot)$ is a balancing function if $g(t) = t\cdot g\Big(\frac{1}{t}\Big)$. In the paper the authors say that $g(t) = \min \{ h_{\theta}(t), t\cdot h_{\theta}(\frac{1}{t}) \}$. Thus, $g$ in this case becomes a balancing function when $h_{\theta}$ is a balancing function. In proposition 2, the authors also say that they can always find a $\tilde{\theta}$ such that $h_{\tilde{\theta}}(t) = l(t)$ ie a balancing function which requires the universal approximation theorem. However, I am not sure how this can be proven for a finite-capacity network which will be used in practice. Thus, if this $h_{\theta}$ is not a balancing function (which may very well happen with a finite capacity net), then the resulting function $g$ is also not balancing function. Thus, what effect does it have on the properties of the sampler? Or am I missing something here?
> >
> > Also, the authors said that the network capacity limits the balancing functions they can represent. What's the trade-off look like here? Is there any experiment that for more complex problem the authors might have to resort to really large nets leading to a huge computational overhead?

---

> > > ### Author Response · Authors · 2021-08-23
> > > **Discussion about Comparison and Proof**
> > >
> > > First of all, thank you for the thoughtful discussion and the willingness to do so. We appreciate the good intent of the reviewer. Please, find below the answers to your questions:
> > >
> > > **Prior work** We said that it is unfair asking for a comparison, simply because at that time we were not checking arXiv. Indeed, the two works have been officialized only recently. However, we are grateful to the reviewer for pointing them out and we acknowledge him/her. We have also provided a detailed comparison in our rebuttal, which clearly define what are the boundaries of the works and the elements of novelties (indeed this needs to be added to the main paper). We take a step further, especially for the comparison with [1], which is mostly related to us. From the table in our rebuttal, we show that [1] is an approximation of [0] using $g(t)=\sqrt{t}$. Note that this approximation comes with a cost of reduced sampling efficiency compared to [0], as quantitatively stated by Theorem 1 in [1] (and also the paragraph precedent to this theorem). Now, we know from our experiments (see for example Figure 3c and 3d) that we have a clear advantage over [0] with $g(t)=\sqrt{t}$, consequently also to [1]. Note also that [1] and our work are orthogonal to each other. In principle, we can apply the same approximation of [1] to our strategy.
> > >
> > > **Universal Approximation** Please, let us provide a proof, as there is a misunderstanding.
> > >
> > > Note that LSB 2 uses the form $g(t)=\min\\{h(t),t h(1/t)\\}$ for all $t>0$ (we remove theta for the sake of simplicity in the notation, without affecting the validity of the proof), where h(t) is our neural net.
> > >
> > > Now, if $h(t)$ is a balancing function, namely $h(t)=t h(1/t)$, then $g(t)=\min\\{h(t),t h(1/t)\\}=\min\\{h(t),h(t)\\}=h(t)$, Therefore, $g(t)$ is also a balancing function.
> > >
> > > If $h(t)$ is not a balancing function, then we can still show that g(t) is balancing. Indeed:
> > >
> > > $$t g(1/t)= t \min\\{ h(1/t), 1/t h(t)\\}=\min\\{ t h(1/t), h(t)\\}=\min\\{h(t),  t h(1/t)\\}=g(t)$$
> > > Thus concluding the proof.
> > >
> > > Regarding the capacity of the network, we agree with the reviewer that there is indeed a tradeoff between expressiveness and computational overhead. However, we want to stress that $g(t)$ is a function mapping from a real line to another real line. Therefore, there is no need for having an overly large network, also because most of the known balancing functions can be well represented by the proposed network. One can also inspect what the network can learn by regressing it to any desired balancing function and then visualizing it.

---

> > > > ### Comment · Reviewer_2Vii · 2021-08-23
> > > > **question on proof**
> > > >
> > > > Thanks for writing the proof formally here since I tried the same thing and I am not sure if it is correct. Specifically:
> > > >
> > > > You use the fact that $t \cdot \min$ {$h(\frac{1}{t}), 1/t h(t)$} = $\min$ {$t \cdot h(\frac{1}{t}), h(t)$}.
> > > >
> > > > However, this is not always true. You can check this for example by taking $h(t) = t^2$ which yields the two sides as follows:
> > > >
> > > > $t \min$ {$\frac{1}{t^2}, t$} = $\min$ {$\frac{1}{t}, t^2$} and here LHS is not equal to RHS.
> > > >
> > > > Is this correct or am I missing something here still?

---

> > > > ### Comment · Reviewer_2Vii · 2021-08-23
> > > > **previous comment**
> > > >
> > > > Sorry, I just realised that my comment worked since you have the condition $t>0$ which I missed earlier.

---

### Official Review · Reviewer_j21t · 2021-07-13

**Rating:** 4
**Confidence:** 4

**Summary:**

This paper improves upon a recently proposed MCMC method ([Zanella. JASA 2020], cited as [28] in the paper) and considers optimizing the choice of balancing function in that method. A collection of rigorous techniques are proposed so that the balancing function can be represented by a neural network, and the network is trained by minimizing a heuristic objective (two possibilities are provided), which seems to be an approximation of the KL divergence from a Metropolis-Hasting proposal to the target distribution. Limited empirical evidence of effectiveness is provided.

**Main Review:**

The method is new and interesting. I especially like results in Section 3.1, which are smart and innovative. Overall, I feel the idea has a lot of potential and deserves a better demonstration, both theoretically and experimentally. More precisely, if I understood correctly, the goal is to accelerate the convergence of Metropolis-Hasting-type MCMC algorithms. Ideally, some theoretical analysis should be provided to quantitative establish an improved convergence speed, which is not the same as showing detailed balance (which is rather standard for Metropolis-Hasting) or ergodicity (which is not quantitative and not hard for finite state space Markov chains). However, I understand that this kind of analysis is typically highly nontrivial, and therefore empirical demonstration is also a possibility. In this case, some convincing numerical experiments with comparison to SOTA methods should be designed and provided. Such comparisons are currently missing, except one that compares against [28], which this work aims at improving. How much improvement over [28] was obtained was not very clear either; for example, I wonder whether Table 2,3 imply significant improvement or marginal ones. At the same time, can effective sample size be used to illustrate (improved) speed of convergence? I also don’t understand what `achieving fast convergence while preserving fast mixing’ means in Section 5.2, as one way to quantify convergence speed is through mixing time. More evidence of improved convergence speed will significantly strengthen the paper and illustrate the advantages of the new idea.

Some additional comments are:

Not all MCMC methods are based on proposals. Perhaps the authors were thinking about Metropolis-Hasting, which is just one class (albeit very important) of MCMC methods. Other classes include, for example, Gibbs sampler, (unadjusted-)LMC, etc. Not all references cited in the section of Related Work, however, seem to be Metropolis-Hasting based approaches.

Do "continuous domain" and "discrete domain" mean continuous state-space and discrete state-space? It has been repeated that less effort has been devoted to the discrete counterpart (of MCMC), but in general finite state-space Markov processes (chains) are easier than the infinite ones (for example, if a finite chain is ergodic then it must be geometrically ergodic, but this is not true for continuous state space). It will be great if more clarification can be provided on what makes sampling from a finite set a challenging problem and why tools developed for general state-space are insufficient/inefficient.

Is an indicator function $1[x’\neq x]$ missing in front of the first term on the right hand side of equation(1)?


**Time Spent Reviewing:**

3

---

> ### Author Response · Authors · 2021-08-10
> **Clarifications on Objective and on the Practical Advantages of Our Strategy**
>
> We thank the reviewer for the time spent to read our paper. We appreciate the fact that you find our idea innovative and rich of potential. Please, see below some considerations to your comments, which helped us to better highlight the strength of our paper.
>
>
> **Clarification on the objective**
>
> “seems to be an approximation of the KL divergence from a Metropolis Hasting proposal to the target distribution”. There is something more than that. Note that minimizing only the KL between the transition probability and the target distribution is equivalent to approximate the expectation over $p$ using an expectation over a uniform distribution. In other words, this is equivalent to use only the first addend in Eq. (6) of our paper. However, our objective is more sophisticated than that, as it attempts to minimize the dependence between consecutive samples, by minimizing an approximation of Eq. (3) (indeed we are approximating $p$ as a mixture between a uniform distribution, namely $Q_{init}$, and a more informative one, namely $Q_{\theta_0}$).
>
> Also, note that a similar heuristic objective has been already proposed in [1] (see the corresponding Eq. (8) in [1]) to automatically configure the hyperparameters of a continuous MCMC sampler. However, the authors in [1] use the squared jump distance in place of our KL. In our case, the squared jump distance is uninformative (recall that in our discrete case the neighborhood is based on the Hamming distance 1 and therefore the squared jump distance is constant for all possible steps). The connection with [1] highlights another important interpretation of our heuristic objective. Indeed, the first addend in our Eq. (6) (equivalently the second addend in Eq. (8) of [1]) encourages fast burn-in, while the second addend in our Eq. (6) encourages fast mixing.
>
> These aspects are necessary to learn a balancing function with optimal behaviour with respect to the target distribution and are part of our contribution.
>
>
> **Improvement over [0]**
>
> We respectfully disagree with you, because the advantage of our strategy is clear compared to [0].
> Firstly, note that the process of selection of the balancing function is automated differently from [0]. Secondly, the learnt balancing function has both good burn-in and mixing properties. This is not the case in [0], as the proposed balancing functions are either good in terms of burn-in or good in terms of mixing, but not both at the same time (see for example Figure 2 in our paper). In our case, we recover balancing functions with a smaller number of burn-in iterations while having good mixing behaviour (see for example Figure 3 and 4 together with Table 2). Overall, the number of evaluations of the target distribution is reduced compared to [0].
>
> **Minor comments**
>
> “Do "continuous domain" and "discrete domain" mean continuous state-space and discrete state-space?...”.
>
> Yes. We understand that sampling on a discrete state-space might seem simpler than sampling on a continuous one and therefore it might be worth looking for strategies developed for a continuous state-space and apply them to the discrete one. While this is a possible and promising direction to tackle the problem, we want to highlight the issues related to this approach. Indeed, note that there is no notion of gradient in the discrete case. This implies that continuous strategies require to learn a mapping to relax the discrete space to a continuous one. This comes with two issues: the first one is to identify a proper mapping enabling a “simplified” sampling (and it is not always trivial to define such mapping respecting the topological properties of the discrete space) and the second one is that such mapping comes with an additional computational cost to transform the discrete-state space in a continuous and "simplified" one and to invert such transformation.
>
>
> “Is an indicator function...”
>
> No, as we are using $N(x)=\\{y: \|y-x\|=1\\}$ as in [0].
>
>
> [0] “Informed proposals for local MCMC in discrete spaces” Zanella 2020.
>
> [1] “Generalizing Hamiltonian Monte Carlo with Neural Networks” by Levy et al. 2018

---

### Official Review · Reviewer_PwEa · 2021-07-20

**Rating:** 5
**Confidence:** 4

**Summary:**

The paper built on top of the work "Informed proposals for local MCMC in discrete spaces" Giacomo Zanella. Author propose to learn the balancing function by optimising mutual information objective between $KL[p(x)T(x'|x) | p(x)p(x')]$.

**Ethical Concerns:**

No ethical concerns to discuss.

**Limitations And Societal Impact:**

No limitations to discuss.

**Main Review:**

Minor comments

L 63 Formal definition of $N(x)$ is missed, I used from  Zanella.
L 71 It is not clear why only acceptance rate important
L 101 In proposition 1 maybe it will be interesting to discuss mixture of balancing functions vs. mixture of transitional kernels with corresponding balancing functions.

General

In Proposition 2 general form of balancing function $\min \left[h(t), t h( \frac{1}{t} )\right]$ was proposed in Zanella. So, contribution in parametrization h with DNN, which is also interesting. Then the crucial question, in my opinion, is the selection of the optimization criterion. This however discussed very briefly in 3.2 In L 144 motivation is not clear "Note that the ideal case would be to sample from $p$ independently", given the fixed neighbourhood of $N(x)$ (see also arXiv 1810.07151, arXiv 1711.09268 as examples of discussion about loss for transition kernel learning). Moreover, given the fixed neighbourhood of $N(x)$, I don't understand L 158 $T(x'|x) = p(x')$. ?!
Despite that I can guess the overall reason for the objective: the balancing function should incorporate global information about the target density. I suppose the choice of the loss should be discussed more.

The next paper is discussed universally optimal balancing functions. First, I don't understand (and paper don't provide) any intuition of existing such thing. For continuous support, such function does not exist, what is different in discrete cases? Assuming that it exists, I also don't understand the selected functions to compare: I can provide infinitely many such functions in terms of hyperbolic functions. Maybe I miss the point.


**Time Spent Reviewing:**

4

---

> ### Author Response · Authors · 2021-08-10
> **Clarifications about Objective and Balancing Functions**
>
> We thank the reviewer for the time spent to read the paper and the comments. Please, find below the answers to your questions.
>
> **Clarifications about the objective**
>
> Our objective attempts to minimize the dependence between consecutive samples, by minimizing an approximation of Eq. (3). Indeed,  we approximate $p$ using a mixture between a uniform distribution, namely $Q_{init}$, and a more informative one, namely $Q_{\theta_0}$. The objective in Eq. (6) can be derived by the linearity of expectation on this mixture. Note that if we would consider only $Q_{init}$, thus discarding $Q_{\theta_0}$, we would end up to compute the KL divergence between the transition probability and the true target, thus discarding the dependence between consecutive samples (as measured by mutual information). However, we can do better by exploiting also the term with $Q_{\theta_0}$, as being more informative than the uniform distribution $Q_{init}$.
>
> Also, note that a similar objective has been already proposed in [1] (see the corresponding Eq. (8) in [1]) to automatically configure the hyperparameters of a continuous MCMC sampler. However, the authors in [1] use the squared jump distance in place of our KL. In our case, the squared jump distance is uninformative (recall that in our discrete case the neighborhood is based on the Hamming distance 1 and therefore the squared jump distance is constant for all possible steps). The connection with [1] highlights another important interpretation of our objective. Indeed, the first addend in our Eq. (6) (equivalently the second addend in Eq. (8) of [1]) encourages fast burn-in, while the second addend in our Eq. (6) encourages fast mixing.
>
>
> **Clarifications about balancing functions**
>
> While we agree with the reviewer that there are infinitely many balancing functions (take for example LSB 1 and you can get an uncountable set), our first parametrization (LSB 1) is general, in the sense that it supports a linear combination of any kind of balancing function. Specifically, given a set of predefined balancing functions, we can take our parametrization to generate new balancing functions. The choice of the four functions is based on [0]. Also, note that our second parametrization (LSB 2) can theoretically represent any balancing function, without requiring to explicitly define the set of balancing functions beforehand (like it is done in LSB 1). Does this address your question?
>
>
> **Minor comments**
>
> L63 indeed we are using the definition in [0].
>
> L71 what is it unclear?
>
> L101 Proposition 1 is valid for any linear combination of positive coefficients. In our case, we used normalized coefficients to softly select one balancing function. However, we are not able to see the connection between “the mixture of balancing functions and the mixture of proposal distributions”. Indeed, note that our proposal can be rewritten as
>
> $$Q(x’|x)=\cdots=\sum_{i}w_i \frac{g_i(p(x’)/p(x))}{\sum_{j}w_j Z_j(x)}1\[x’\in N(x)\]$$
>
> while the mixture of proposals would be
> $$\tilde{Q}(x’|x)=\sum_{i}w_i\frac{g_i(p(x’)/p(x))}{Z_i(x)}1\[x’\in N(x)\]$$
>
>
> In general, the two are not equivalent. Does that answer your question?
>
>
> [0] “Informed proposals for local MCMC in discrete spaces” Zanella 2020.

---

> > ### Comment · Reviewer_PwEa · 2021-09-01
> > **Comments**
> >
> > L71 Because maximising only acceptance rate could suppress exploration (that being said your objective has both terms for encouraging mixing and AR)
> > L101 I understand that they are different. Hence, it is not clear what to prefer.
> >
> > Clarifications about the objective
> > Thank you for comments. While I understand the motivation, it is a bit handwaving and doesn't answer on question while this objective is better than other. For example one can consider symmetric version of variational representation of mutual information. I hoped to see some connection to the balancing functions. For now it is looked for me like general choice for any proposal distribution. While this is reasonable choice in practice, I believe it should be a bit more justified.
> >
> > Clarifications about balancing functions
> > + "The choice of the four functions is based on [0]."  For my point of view, it is not the valid argument, as it doesn't provide information.
> >
> > I suppose the paper rises an interesting question and provide some solution. However, I believe choices in algorithms should be more justified theoretically (as possible) or at least empirically.  Hence, I will keep my score.

---

> > > ### Author Response · Authors · 2021-09-02
> > > **Clarifying Misunderstandings/Discussion of Papers Related to Objective/Clarifications about Balancing Functions**
> > >
> > > We thank the reviewer for the reply. Please, find below some comments which strengthen the novelty of our objective and clarify the motivation on the choice of balancing functions for the first parametrization LSB 1.
> > >
> > > **Discussion of papers related to objective**
> > >
> > > Regarding the objective, the reviewer is concerned about its comparison against two existing works.
> > >
> > > [1] “Generalizing Hamiltonian Monte Carlo with Neural Networks” by Levy et al. 2018 (ICLR)
> > >
> > > [2] “Metropolis-Hastings view on variational inference and adversarial training” by Neklyudov et al. 2018 (arXiv)
> > >
> > > Regarding [1], the paper is already cited in our manuscript. Furthermore, we have also discussed in the rebuttal the main similarities and differences in terms of objective with [1]. However, the objective in [1] is based on the squared jump distance and it cannot be applied to our case (please refer to the first answer in the rebuttal, in the first answer we forgot to refer to what [1] means).
> > >
> > > Regarding [2], we realize that we haven’t discussed it in the rebuttal. However, please find here a summary of the paper followed by the discussion about the main differences. Note that the authors of that paper consider independent proposals, namely $q(x’|x)=q(x’)$. “An attractive property of independent proposals is their ability to make large jumps, and if this can be done while keeping the acceptance rate high, the autocorrelation of the chain will be small.” Therefore, the goal of the authors is to learn an independent proposal by maximizing the acceptance rate, namely $AR=\min\\{1,\frac{p(x’)q(x)}{p(x)q(x’)}\\}$. In practice, the authors use a surrogate lower bound on the acceptance rate, namely $$AR \geq 1-\sqrt{1/2KL(q(x)\|p(x))+1/2KL(p(x)\|q(x))}$$
> > >
> > > *Differences with [2]*:
> > >
> > > 1. We are not considering independent proposals, rather locally balanced proposals. Hence, the argument about maximizing the acceptance rate does not hold in our case.
> > > 2. We are maximizing a proxy for the mutual information (please refer to the first answer in the rebuttal for recalling details), and not a proxy for the acceptance rate. The two are clearly different, see our eq. (5), where it is true that the acceptance function influences the objective but it’s not the only term, and also eq. (6), which enforces fast burn-in and fast mixing (again refer to our first answer in the rebuttal for this interpretation).
> > > 3. (Less importantly) We are considering sampling in the discrete domain, rather than in the continuous one like in the mentioned paper.
> > > 4. (Less importantly) We are considering locally balanced proposals and parameterise only the balancing functions. Whereas, the authors parameterize the whole proposal using a neural network.
> > >
> > > That said, we thank the reviewer for pointing out [2] and we can add the discussion about the objective in the related work. However, we want to point out that the contribution of this work consists of two parametrizations (LSB1 and LSB2 for locally balanced proposals) and an objective function to learn such parametrizations. These are undeniable novelties to improve over the recent work of Zanella [3].
> > >
> > > **Clarifications about balancing functions**
> > >
> > > The choice of the four balancing functions for LSB 1 has been dictated by the ones proposed in the work of Zanella [3]. Note that the result of the parametrization (LSB 1) is valid independently from the choice of the balancing function. Furthermore, note that our aim is beyond the selection of balancing functions. Indeed, we want to directly learn such functions without selecting them, as in our parametrization LSB 2.
> > >
> > > **Minor**
> > >
> > > L71 We simply recall some properties of balancing functions from the work of Zanella [3]. Let us explain a bit better what we meant in the paper. The acceptance function can be rewritten in simpler form thanks to the balancing property. Furthermore, the acceptance function depends only from $Z(x)$ and $Z(x’)$, roughly computing “the mass in the neighborhood of $x$ and $x’$”, respectively. Since we are dealing with scores computed on neighborhoods and also because the two neighborhoods on $x$ and $x’$ are highly overlapping (due to the local step), the ratio $\frac{Z(x)}{Z(x’)}$ and consequently the acceptance function tends to be higher compared to the case of a non-balancing proposal (recall that in this case the acceptance function depends on the pointwise values of the proposals and that while there might be a large difference between pointwise values, the scores on the neighborhoods might be more balanced). Obviously, the ratio $\frac{Z(x)}{Z(x’)}$ depends on the balancing function itself and therefore it could be affected by its choice. There is a more thorough discussion and experimental evidence supporting this fact in the main paper of Zanella [3].
> > >
> > > [3] “Informed proposals for local MCMC in discrete spaces” Zanella 2020.

---

### Author Response · Authors · 2021-08-22
**General Request**

Dear Reviewers,

thank you for the feedbacks to our paper.
We are writing to ask to take into consideration our answers in your final judgement.

Thank you for the attention.
Kind Regards,
Authors

---

### Decision · Program_Chairs · 2021-09-27

**Decision:**

Reject

**Comment:**

This paper considers the learning to accelerate the MCMC sampling. Specifically, the authors proposed a learning objective, which is minimizing the dependence between consecutive samples, for learning the proposal. Meanwhile, the authors discussed the parametrization of the balancing function. The authors evaluated the proposed algorithms empirically on 2D Ising model and Bayesian networks.

All the reviewers think the paper is interesting and promising. However, there are still several concerns:

- The paper is largely relying on [Zenella, 2020], and shared some similarity with the related work as Reviewer PwEa, 2Vii mentioned [1, 2, 3]. The significance of the paper is not clear for current version.

- The choice of the loss should be discussed more as Reviewer PwEa and j21t raised.

- The empirical experiments is not comprehensive (Reviewer j21t) and experimental analysis around the method is lacking (Reviewer 2Vii).

In sum, the paper indeed is proposing some interesting method, however, the current version is not ready to be published. I encourage the authors can take the suggestion from the reviewers into account to improve the draft and submit to next venue.

[1] "Learning Discrete Energy-based Models via Auxiliary-variable Local Exploration" by Dai et. al 2020

[2] "Oops I Took A Gradient: Scalable Sampling for Discrete Distributions" by Gratwohl et.al 2021

[3] "Sampling in Combinatorial Spaces using SurVAE Flow Augmented MCMC" by Jaini et.al 2021